# Specific ion effects at graphitic interfaces

Cheng Zhan[1], Maira R. Cerón [1], Steven A. Hawks [1], Minoru Otani [2], Brandon C. Wood[1], Tuan Anh Pham[1]*, Michael Stadermann[1]* & Patrick G. Campbell [1]*

Improved understanding of aqueous solutions at graphitic interfaces is critical for energy storage and water desalination. However, many mechanistic details remain unclear, including how interfacial structure and response are dictated by intrinsic properties of solvated ions under applied voltage. In this work, we combine hybrid first-principles/continuum simulations with electrochemical measurements to investigate adsorption of several alkali-metal cations at the interface with graphene and within graphene slit-pores. We confirm that adsorption energy increases with ionic radius, while being highly dependent on the pore size. In addition, in contrast with conventional electrochemical models, we find that interfacial charge transfer contributes non-negligibly to this interaction and can be further enhanced by confinement. We conclude that the measured interfacial capacitance trends result from a complex interplay between voltage, confinement, and specific ion effects-including ion hydration and charge transfer.

---

[1] Lawrence Livermore National Laboratory, Livermore, CA 94550, USA. [2] National Institute of Advanced Industrial Science and Technology (AIST), 1-1-1 Umezono, Tsukuba 305-8568, Japan. *email: pham16@llnl.gov; stadermann2@llnl.gov; campbell82@llnl.gov

Aqueous solutions at voltage-biased graphitic interfaces are essential to a variety of energy and environmental technologies, including supercapacitors for energy storage[1,2] and capacitive deionization for water desalination and purification[3–6]. Within these devices, a detailed understanding of specific ion effects at the interface and their influence on interfacial capacitance is crucial for predicting and optimizing performance. However, a general consensus on how the interfacial response is governed by intrinsic properties of ions remains largely lacking, and contradictory trends have been reported, even for simple alkali-metal ions. For instance, measurements of activated porous carbons in aqueous solutions showed that the overall capacitance increases with the ionic radius of the ions according to $Li^+ < Na^+ < K^+$[7–9], whereas an inverse relationship was reported by another experimental study[10]. These contradictory results highlight that much is left to be understood regarding ion effects at graphitic interfaces. In addition, they point to the need for a better understanding of how these effects are influenced by confinement, which necessarily accompanies the porous nature of many carbon materials.

In a recent study, highly ordered pyrolytic graphite (HOPG) was employed as a model system for investigating specific ion effects on graphene capacitance[11]. By considering solutions with a series of alkali-metal cations, from $Li^+$, $Na^+$, $K^+$, $Rb^+$ to $Cs^+$, with $Cl^-$ counterions, the authors showed that the basal-plane capacitance of the system increases with the ionic radius of the cations, with CsCl yielding the highest capacitance among the solutions. This study provides the first experimental demonstration of the relationship between cation identity and capacitance for graphene as a well-defined carbon electrode in contrast with activated carbons, for which definitive conclusions are generally more difficult to draw due to the inherent chemical and structural complexity.

In addition to experimental studies, molecular dynamics (MD) simulations with classical force fields have been extensively employed to investigate aqueous solutions near graphene[7,12–14], as well as within carbon nanotubes[15–23] and carbon slit pore electrodes[24,25]. While these simulations have significantly advanced our understanding of the electric double layer (EDL) at graphitic interfaces, notable discrepancies between theoretical and experimental studies remain to be addressed. As a prime example, Bo et al.[7] calculated the capacitance of a graphene electrode in aqueous electrolytes with alkali-metal cations using classical MD simulations, showing that the capacitance does not depend on the cation type due to the high dielectric constant of liquid water. This is, however, in contrast to the experimental results reported in the same study. To explain this discrepancy, a kinetic-dominated charging mechanism was proposed and supported by capacitance measurements at different scan rates, pointing to the importance of ion kinetics on the electrochemical performance. However, electrochemical impedance spectroscopy measurements in ref. [11] show that distinctive difference in the capacitance between the cations remains at a low frequency; for instance, the measured capacitance in KCl and LiCl solutions were $\sim 8\,\mu F\,cm^{-2}$ and $\sim 5\,\mu F\,cm^{-2}$, respectively, resulting in a $C_{K^+}/C_{Li^+}$ ratio as large as 1.6. This indicates that ion kinetics may not be solely responsible for the difference in the capacitance between different aqueous solutions, motivating further development in theoretical descriptions of the electrical response of solutions at graphitic interfaces.

Here we combine hybrid first-principles/continuum simulations with electrochemical measurements to understand ion effects at graphitic interfaces. Rather than relying on conventional computational methods grounded exclusively in classical force fields, for which results depend on the specific parameterization, we utilize a newly developed theoretical framework that combines first-principles density functional theory (DFT) with the reference interaction site model (RISM) to explicitly account for electronic effects in the description of the ion–electrode interaction[26]. We show that large polarized cations exhibit strong adsorption and a significant degree of charge transfer at the interface with graphene. In addition, these effects are found to be enhanced for solutions confined in graphene slit pores. Combining these theoretical findings with direct electrochemical experiments enables us to elucidate key factors that determine the structure and electrical response of aqueous solutions at the interfaces and at the same time provide insights into the fundamental interplay between confinement and specific ion effects on both capacitance and ion selectivity of porous carbon materials.

## Results

**Cation adsorption at the graphene interface**. We begin by discussing the behavior of aqueous LiCl, NaCl, KCl, and CsCl solutions near an ideal graphene electrode, which represents a well-defined model system that can be reasonably compared with the recent experimental investigation on HOPG[11]. We first compute the capacitance of these systems using a purely classical description for the solution environment within the RISM approach (see "Methods"), with the electrode explicitly described by DFT. As shown in Supplementary Fig. 1, we find that all the solutions exhibit nearly identical theoretical capacitance. This behavior is similar to the previous MD studies[7]. However, it is not consistent with experimental measurements of the graphene basal plane reported in ref. [11], which show that the capacitance should increase with the ionic radius. This indicates that classical descriptions of the ions based on the point charge approximation —while likely adequate for bulk electrolyte solutions—are insufficient for properly capturing the experimentally measured cation effects at interfaces.

Next, to seek a more realistic description of the cations at the interface and elucidate the origins of the discrepancies with experiments, we include an explicit cation in each of our computational models of the interface and directly investigate its adsorption behavior. In this case, the explicit cation and carbon electrode are treated by DFT, and the rest of the electrolyte is described through the RISM approach. The calculated potential energy surface (PES) as a function of the distance between the cation and a charged graphene sheet (with a charge density of $-0.0052e$ per carbon atom, or about $-0.25\,V$ versus the potential of zero charge), is shown in Fig. 1a. The results, which are referenced to the potential of the ions in the bulk electrolyte solution, indicate that cations with larger ionic radius exhibit stronger adsorption at the interface. In particular, at this electrode charge density, $Cs^+$ and $K^+$ are favorably absorbed at the interface, exhibited by a negative local minimum in the free energy landscape at a distance of 3.3 and 3.0 Å from the graphene electrode, respectively. This is in contrast to the smaller $Li^+$ and $Na^+$ ions, which exhibit unfavorable interfacial adsorption at this charge density. Nevertheless, adsorption of these smaller cations can be induced by further increasing the electrode charge density, eventually leading to binding for $Li^+$ and $Na^+$ at 2.3 and 2.7 Å from the graphene plane, respectively (see Supplementary Fig. 2 and Supplementary Table 1). Note that the positions of the first local minima in the PES are highly correlated with the ionic radius.

The conclusion that cations with a larger ionic radius exhibits stronger adsorption at the interface with graphene is also consistent with a recent theoretical study using a combination of DFT and the conductor-like polarizable continuum model (CPCM) for liquid water[13]. However, it is necessary to point out that the absolute binding energy obtained for $Li^+$, $Na^+$, and $K^+$

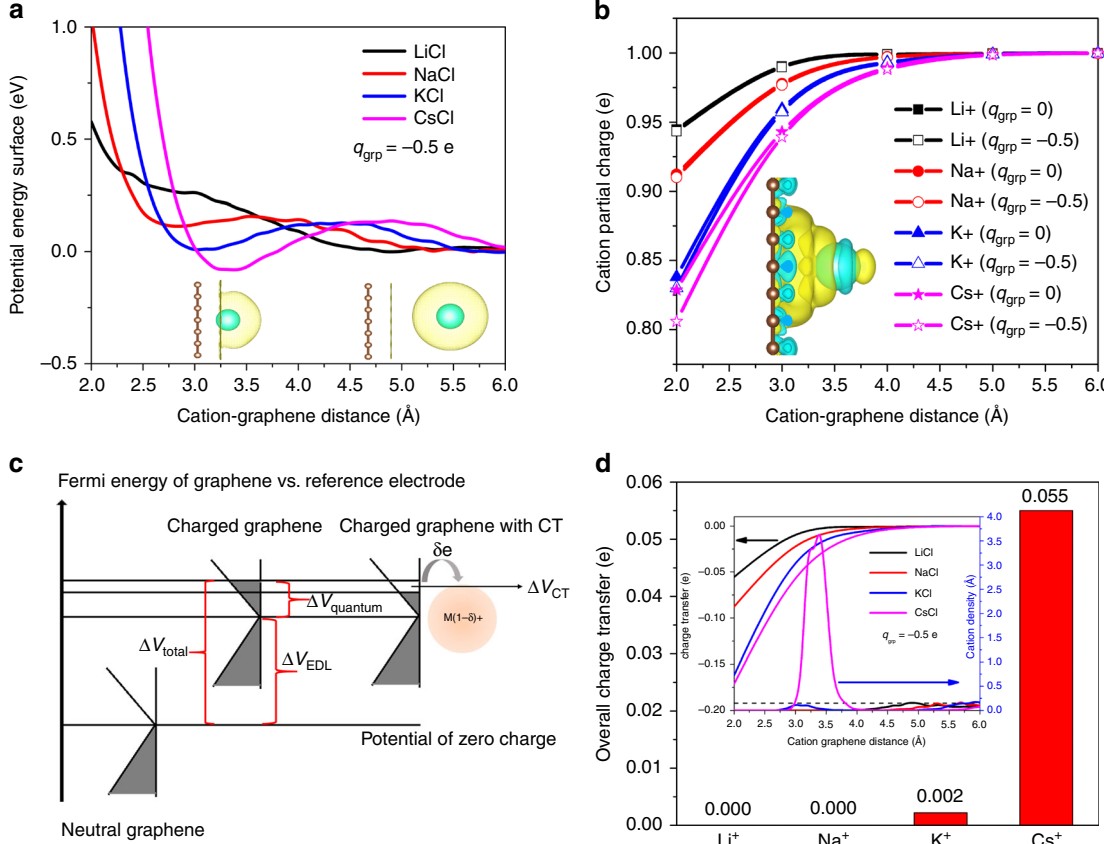

**Fig. 1** Interaction between alkali-metal cations with graphene. **a** Potential energy surface of the cation adsorption on the graphene surface with a total excess charge of $q_{grp} = -0.5\,e$ in 1 M LiCl, NaCl, KCl, and CsCl aqueous electrolytes. Inset represents the cation–graphene system, where the green-yellow circles represent a hydrated alkali cation. **b** Ionic charge of the cations at different $q_{grp}$ and adsorption locations. Inset: interfacial charge transfer between Cs$^+$ and the graphene electrode as an example, where the cyan and yellow isosurfaces indicate charge depletion and accumulation regions, respectively. **c** Schematic description of the charge transfer (CT) effect on the potential response of charged graphene electrode. **d** The overall charge transfer obtained by integrating the charge transfer per ion, shown in **b**, and local cation density (inset). The isosurface is set to be $10^{-4}$ and $1.6 \times 10^{-4}$ a.u. for the oxygen density in the ion solvation shell (in **a**) and electronic density difference (in **b**), respectively

on graphene is weaker in the current study compared to that of ref. [13], which can be attributed to several differences in the computational set-up and solvation models. In particular, in ref. [13], the cation adsorption energy is calculated for pure water described by the CPCM model, whereas our calculations were carried out for 1 M electrolytes. As we show in Supplementary Fig. 3, the inclusion of ions in our simulation weakens the interaction between a charged graphene and cations. In addition, in variation with the CPCM model, the RISM approach employed here allows for the description of the local solvent structure around the ions, as illustrated in the inset of Fig. 1a. In this regard, the RISM approach provides a more realistic description of the desolvation process of ions when they approach the graphene surface, which introduces an additional energy penalty for ion adsorption.

The adsorption ordering of the cations shown in Fig. 1a can also be directly related to their hydration energy. Specifically, larger ions, such as Cs$^+$, yield a weak solvation shell[27] and therefore can be easily desolvated and adsorbed at the electrode interface (see inset Fig. 1a). In contrast, the smaller ions have a much stronger solvation environment and prefer to remain solvated, preventing them from approaching the graphene interface. Adsorption occurs only at more negative voltages, for which the cation attraction becomes sufficient to overcome the solvation energy. Notably, this implies that the adsorption response—and hence the interfacial capacitance—for Li$^+$ and

Na$^+$ should depend qualitatively on the applied voltage. Accordingly, we conclude that the structure of aqueous solutions at the interface with graphene is governed by both ionic radius and hydration strength, as well as applied potential, which alters the relative competition between these two factors.

Nevertheless, the aforementioned results are insufficient for fully describing the experimental trends of increasing capacitance with increasing ionic radius[11]. For instance, they would predict that the larger Cs$^+$ ion should sit at a longer distance from the graphene plane than Li$^+$ and therefore produce a weaker capacitance, yet the opposite trend is seen. To explain this, we proceed to investigate other electronic effects at the interface, including charge transfer, which are enabled by the inclusion of an explicit ion in our computational model. As shown in Fig. 1b, Bader charge analysis[28] of the explicit ions in the interfacial region indicates that they generally exhibit a smaller ionic charge compared to the bulk values. This behavior is found to stem from the charge transfer from graphene to the ions, which becomes more significant at short graphene–ion distances. We find that the deviation of the ionic charges from their bulk values is more significant for more polarizable ions, particularly K$^+$ and Cs$^+$. For instance, at a distance of around 3.0 Å from graphene, Cs$^+$ yields a ionic charge of 0.93$e$, as compared to a corresponding value of 0.98$e$ found for Li$^+$. These results indicate that charge transfer is not negligible when the cations are within 4 Å from graphene (i.e., unsolvated and specifically adsorbed) and reinforce

the conclusion that the point-charge approximation for the cations is not sufficient for describing ion impacts on graphene capacitance in aqueous solutions.

The analysis of ion adsorption and charge transfer have important implications for understanding cation effects on the measured capacitance of graphene. As illustrated schematically in Fig. 1c, the overall potential response $\Delta V_{\text{total}}$ of a graphene–liquid interface with excess charge is related to the Fermi level shift that occurs upon the filling of empty electronic states of the electrode. When charge transfer from graphene to the cations is taken into account, not all of the excess charge is diverted toward filling the electrode electronic states; rather, a part of it is transferred to the electrolyte, leading to a smaller $\Delta V_{\text{total}}$. This implies that electrolytes that experience stronger charge transfer with graphene would experience less voltage drop across the interface and therefore yield a larger capacitance. For instance, we find a Fermi level shift in graphene (labeled $\Delta V_{\text{CT}}$ in Fig. 1c) of 0.27, 0.31, 0.40, and 0.42 eV for $Li^+$, $Na^+$, $K^+$, and $Cs^+$, respectively, when the ions approach the interface from a distance of 6 to 2 Å. We emphasize that the spillover of charge to the adsorbed ion upon crossing the intrinsic acceptor level of the ion derives from the low density of states of graphene (also referred to as the "quantum capacitance"), which causes unusually large Fermi level shifts for relatively modest degrees of excess charge.

A more explicit demonstration of the relationship between the degree of charge transfer and the cation adsorption strength is illustrated in Fig. 1d, where the overall charge transfer is computed by integrating the single-ion charge transfer from Fig. 1b with the local cation density (Fig. 1d inset) derived via a Boltzmann weight of the PES from Fig. 1a. It is evident that charge transfer is the most significant for $Cs^+$, followed by $K^+$, implying that graphene would yield a higher capacitance in CsCl and KCl solutions, with especially dramatic enhancements for the CsCl case. This conclusion resolves the remaining inconsistencies with ref. [11], where the graphene capacitance is shown to increase with the ionic radius of the alkali-metal cations.

**Cation intercalation into graphene slit pores.** Having established a detailed understanding of ion effects at the graphene interface, we turn to elucidate the role of confinement by investigating the energy of cation intercalation into graphene slit pores with edges terminated by hydrogen atoms. Here the width of the pore is chosen to be 6 Å, which represents a strong confinement limit and is comparable to the smallest pores measured in activated carbons[29–31]. Similar to our calculations for the graphene interface, the electrode and an explicit cation are treated with DFT, whereas the solvent is described by the RISM approach. The calculated PES for ion intercalation into a slit pore with an excess charge of $-1e$, shown in Fig. 2a, indicates that confinement leads to significant ion selectivity. In particular, we find that the smaller ions, $Li^+$ and $Na^+$, yield an energy barrier of around 0.2 eV to reach the slit pore entrance (indicated by the dashed line in Fig. 2a), which is noticeably larger than the corresponding value of around 0.05 eV found for $K^+$ and $Cs^+$. This corresponds to a difference in intercalation kinetics, which can be again related to the variation in the hydration strength between the cations. In particular, larger ions require less energy to disturb the solvation environment, easing intercalation and further highlighting the importance of ion hydration on selectivity under confinement.

Inside the slit pore, we find that the cations exhibit adsorption behavior that is very similar to that of the graphene interface described in Fig. 1. Specifically, $Li^+$ and $Na^+$ reside closer to the surface, whereas $K^+$ and $Cs^+$ remain in the middle of the slit pore due to their larger ionic radius (Fig. 2b). We also find that larger ions yield a stronger adsorption energy; for instance, $Cs^+$ yields a

binding energy of $-0.35$ eV inside the slit pore, whereas $Li^+$ does not exhibit favorable adsorption. In addition, the ionic charge of cations inside the pore is found to be smaller than the bulk values, yielding a value of $0.961e$, $0.953e$, $0.918e$, and $0.883e$ for $Li^+$, $Na^+$, $K^+$, and $Cs^+$, respectively, as compared to the corresponding value of $0.96e$, $0.963e$, $0.959e$, and $0.942e$ obtained for the flat interface at the ion adsorption position. This indicates a degree of charge transfer from graphene to the ions that exceeds the flat interface case from Fig. 1, except for the smallest ion, i.e., $Li^+$. Two factors contribute to this relatively large reduction in the expected ionic charge, particularly for $K^+$ and $Cs^+$: first, the charge-transfer effects described in the paragraphs above; and second, further stripping of the water solvation shell upon confinement, which reduces the screening between the ions and electrode even more (Supplementary Fig. 4). These results indicate that confinement acts cooperatively to further facilitate and enhance the interfacial response of graphene to specific ions.

**Comparison with cyclic voltammetric (CV) experiments.** In order to compare and validate theoretical results on slit pores, we carry out capacitance measurements of hierarchical carbon aerogel monoliths (HCAMs) in the aqueous solutions. These materials exhibit excellent tunability of the pore size distribution, high chemical stability, and specific surface area and have demonstrated suitability for supercapacitor and water desalination applications[32]. For the current study, we employ HCAMs with dominant pore sizes in the range of 5–6 Å (see Supplementary Fig. 5), comparable to our theoretical models. CV measurements of the HCAM electrodes at two different solution concentrations of 50 mM and 1 M (Supplementary Fig. 6) show that the capacitance follows the order $Cs^+ > K^+ > Na^+ > Li^+$ in aqueous solutions, with more a pronounced trend at the higher concentration. The trend in the measured gravimetric capacitance, summarized in Fig. 3a, is consistent with not only our theoretical predictions but also with previous experimental studies[7–9]. In particular, it is shown that the capacitance increases with the ionic radius and the adsorption energy of the ions at the interface with graphene; however, it decreases with their hydration radius (see Supplementary Table 1). Notably, it is also similar to that observed for the graphene basal plane[11], even though the morphology of the HCAM electrode material is far more complex.

As summarized in Fig. 3b–d, our calculations show that confinement in 6 Å slit pore greatly enhances the degree of charge transfer per ion by almost a factor of two for the large cations (Fig. 3b) compared to that at the flat graphene interface. In addition, as indicated by the calculated binding energies and intercalation barriers, shown in Fig. 3c, the accessibility of the ion into the graphene slit pores increases with ion size, which is a function of the strength of the ion hydration that must be overcome to achieve intercalation. To further illustrate the evolution from strongly confining to weakly confining pores and its effect on binding energy, additional calculations in Fig. 3d are shown as a function of pore diameter. The calculations show that the binding energy of the cations with graphene slit pores always increases with the ionic size, except for the smallest pore size of 5 Å where $Cs^+$ exhibits an unfavorable binding energy (Supplementary Fig. 7) due to its large ionic radius that competes with the pore size dimension itself. We expect that these confinement-derived enhancements also contribute to our measured capacitance, further amplifying the ion-dependent trends. It is also necessary to point out that the porosimetric measurements performed using gas sorption on dry HCAM electrodes indicate that the pore size is dominated by 5 Å pores (gray line in Fig. 3d and black line in Supplementary Fig. 5),

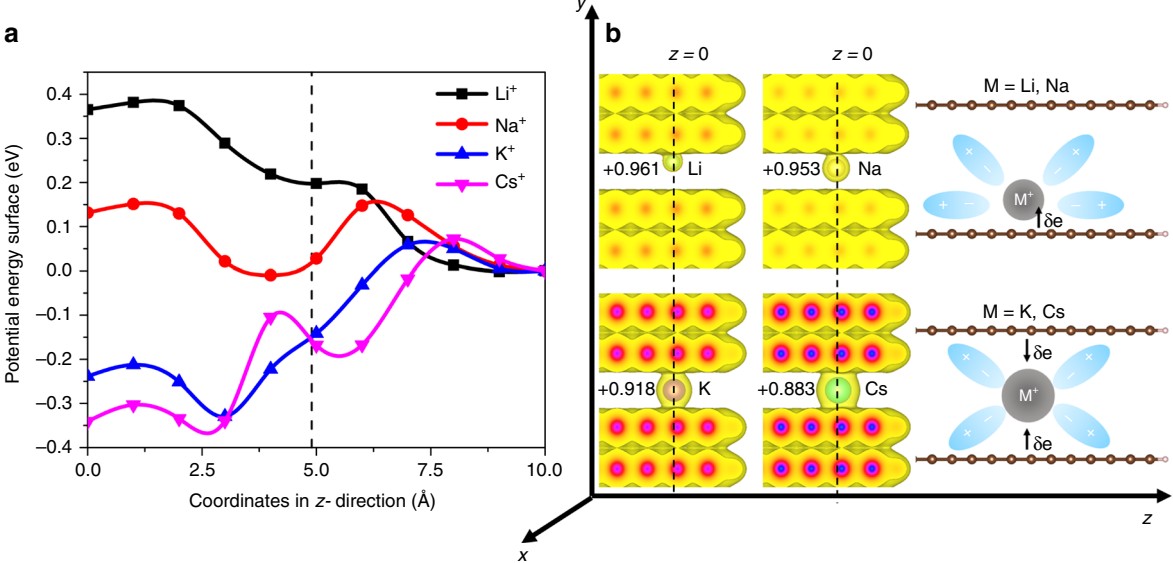

**Fig. 2** Intercalation of alkali-metal cations into graphene slit pores. **a** Potential energy surface computed for the intercalation of the Li$^+$, Na$^+$, K$^+$, and Cs$^+$ ions into a graphene slit pore with a pore diameter of $d = 6$ Å and an excess charge of $q_{grp} = -1e$. The pore is at left, the bulk solution is at right, and the dashed line indicates the location of edge hydrogen atoms at the pore opening. **b** Computed ionic charges and electronic density of the cation-intercalated graphene slit pores with the isosurface value set to $5 \times 10^{-3}$ a.u., and schematic description of confinement effects on the cation adsorption position and charge transfer

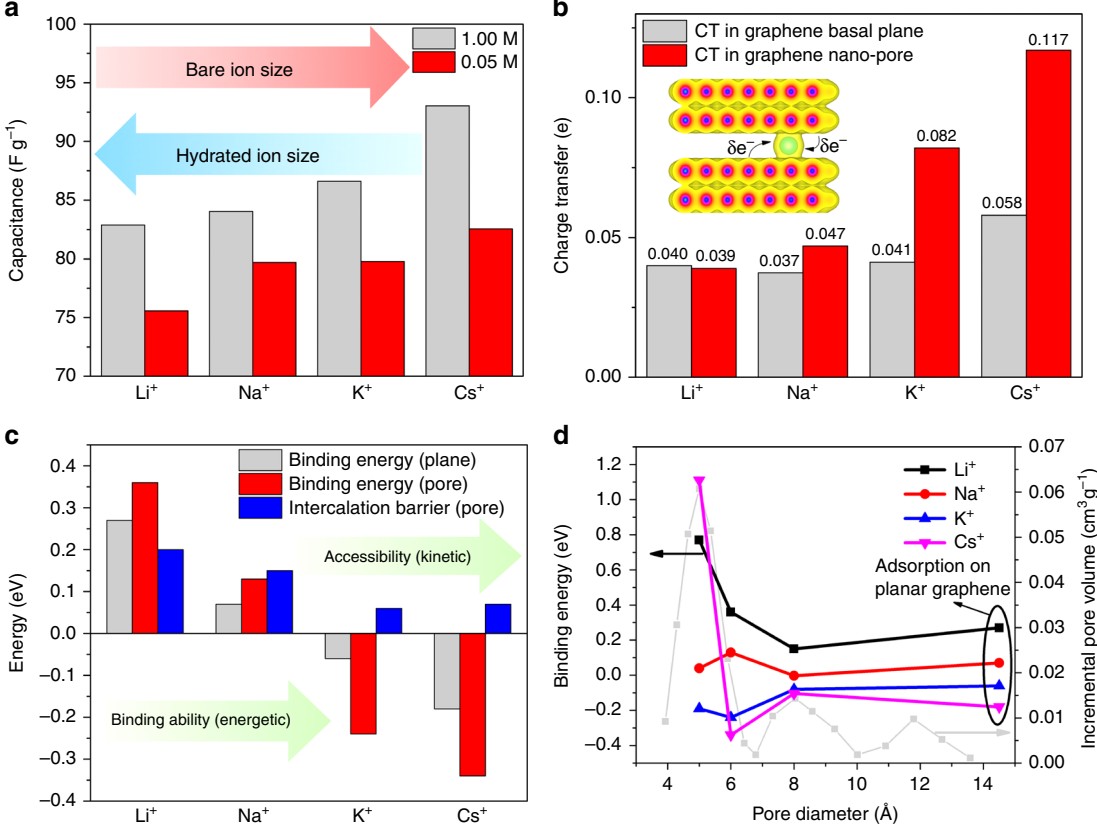

**Fig. 3** Relationship between capacitance, hydration energy, charge transfer, and ion accessibility. **a** Specific capacitance of hierarchical carbon aerogel monoliths (HCAMs) measured in 1 and 0.05 M aqueous solutions at a scan rate of 0.5 mVs$^{-1}$. **b** Charge transfer (CT) per cation on graphene basal plane and in slit pore with a diameter of 6 Å. **c** Binding energies and kinetic barrier for ion adsorption and intercalation computed for the 6 Å diameter graphene slit pore and basal plane. **d** Computed binding energy of the cations as a function of pore size, the experimental pore size distribution is indicated by gray line. In all cases, the excess charge of the electrode is set to $q_{grp} = -1e$, and negative binding energy indicates favorable adsorption

which would seem to limit $Cs^+$ adsorption and insertion according to the predictions. However, during CV cycling, the wetted electrodes are clearly capable of ion adsorption and exhibit large capacitance, as predicted for slightly larger pore geometries. This suggests a possible increase in the pore size upon wetting under operation or that the true pore size of the dry HCAM electrodes is slightly >5 Å due to the uncertainty introduced by the models employed for the determining of pore size distribution (see Supplementary Fig. 5).

Note that our conclusions that ion desolvation due to confinement increases capacitance reflect similar behavior observed for porous materials in solvent-free ionic liquids[2,33,34]. However, we show that the enhanced capacitance is not only related to the reduced electrode–ion distance effect that is often discussed in the literature but also due to the improvement in the charge transfer under confinement. In this regard, our calculations indicate that quantum-mechanical treatment of the interfacial charge transfer is critical for a proper description of the EDL response. This points to the need to go beyond a simple point-charge approximation that is often employed in conventional MD simulations to capture electronic effects within the EDL. In the context of energy storage applications such as supercapacitors, our results also indicate that the choice of solutions with highly polarizable ions would benefit the overall capacitance.

Beyond a complex interplay between ion effects and confinement on the EDL and electrical response, our calculations also provide several key conclusions for ion transport and selectivity[35]. We show that the energy barrier for ion transport into nanopores depends not only on the pore size but also on ion hydration properties[36]. In particular, transport of small ions with large hydration energy, such as $Li^+$, is generally more energetically expensive than for larger, less strongly hydrated ions, such as $Cs^+$. However, our calculations shown in Fig. 3d also point to the importance of the size exclusion mechanism in ion selectivity. In particular, for slit pores with a pore size of <5 Å, $Cs^+$ exhibits a significantly higher barrier for transport compared to other ions due to the larger ionic size compared to others. These results have important implications in tuning morphology of porous materials for selective contaminant removal from water, for instance.

As a final note, it is important to comment on the role of ion dynamics, which is not considered in our simulations, in determining the capacitance of the graphitic electrodes. Along this direction, by using a combination of classical MD simulations and electrochemical measurements, Bo et al.[7] concluded that the charging mechanism and capacitance of graphene in aqueous solutions with alkali-metal ions is dominated by the ion kinetics. Here, to elucidate the role of ion kinetics, we carried out capacitance measurements for different scan rates, and we find that the overall behavior of the capacitance is consistent with the results presented in ref. [7]. In particular, we find that the capacitance of the HCAMs electrode in aqueous solutions decreases with increasing the scan rate and that the capacitance loss is more significant for ions with a smaller ionic radius (see Supplementary Figs. 8 and 9 and Supplementary Tables 2 and 3). For instance, we find that the capacitance decrease of LiCl is higher than CsCl with increased scan rates, which can be attributed to a slower diffusion of $Li^+$ compared to $Cs^+$. These results indicate that, together with specific ion effects and confinement, ion dynamics may play additional roles in determining the electrochemical performances of graphitic electrodes in aqueous solutions. Besides this, ion transport in porous carbons can be affected by the deviation of the electrode material from the ideal graphene slit pore due to, e.g., a complex percolating pore network with different pore sizes[37]. Finally, the presence of functional groups or defects in the electrode may also introduce additional effects on the capacitive performance.

## Discussion

To conclude, investigations of ion hydration, charge transfer, ion adsorption, and intercalation, as well as confinement effects, collectively indicate that solutions with larger cations, such as $Cs^+$, would yield a higher capacitance. This behavior, which is further validated by our experimental data, is the result of an interplay between these various factors, which together form the basis of the relationship between electrolyte composition, confinement, and the measured capacitance. Although our studies were carried out for carbon materials and aqueous solutions, the findings presented here, particularly on specific ion effects, charge transfer, and their interplay with confinement, are likely to hold for other systems with different electrolytes, such as ionic liquids and organic electrolytes, and non-graphitic electrodes, such as low-dimensional systems, e.g., MXene and $MoS_2$[38–40]. Finally, we note that the computational strategy based on the DFT-RISM approach presented in this work is general and applicable to other electrochemical interfaces.

## Methods

**DFT and implicit solvent calculations**. Our computational methodology of aqueous solutions at graphitic interfaces relies on a newly developed approach that combines DFT with the effective screening medium technique[41–43] and implicit solvation model described in the framework of the reference interaction site method (RISM)[26,44,45]. While the DFT-RISM method has been employed to investigate the behavior of solvents at relatively simple solid–liquid interfaces[26], our study presents the first application of the approach to study solutions confined in nanopores, as well as the adsorption of an explicit ion at graphitic interfaces.

A schematic description of our hybrid quantum-continuum approach is shown in Supplementary Fig. 10. In particular, the explicit cations and carbon electrodes were described at the DFT level of theory, whereas the electrolyte was represented by an implicit solvent model consisting of water molecules and 1 M of salts at a temperature of 330 K. Atomic charges and Lennard–Jones (LJ) potentials of the solvent and ions are described through the optimized potentials for liquid simulations all-atom force fields[46,47]. The RISM calculations were performed with the closure model of Kovalenko and Hirata[48], with a cutoff energy of 300 Ry for the solvent correlation function. Our calculations of cations on charged electrodes were carried out by fixing the net charge of the explicit quantum-mechanical region, which is balanced by classical ions in the RISM solvation, and therefore the whole system is neutral. We stress that, in variation with classical MD simulations that are often used to investigate solid/liquid interfaces, our calculations rely on a solvation model for the description of aqueous solutions. In this regard, dynamical effects are not considered in this work, and technical aspects such as the equilibration time in MD simulations are not relevant.

The DFT calculations of the explicit ions and carbon electrodes were carried out with the vdW-DF1 functional[49], and ultrasoft pseudopotentials were employed for the description of the interaction between valence electrons and ionic cores[50]. The electronic wave function and charge density were expanded in a plane-wave basis set truncated at the cutoff energies of 40 and 320 Ry, respectively. For the calculations of cations at the interface with graphene, we model the electrode by a 96-atom orthorhombic cell with lateral dimensions of 14.76 and 17.04 Å. For the calculations of ion intercalation, the slit pore was modeled using a four-layer zigzag graphene nano-ribbon in a $12.81 \times 16.20 \times 45.0$ Å orthorhombic cell consisting of 336 carbon atoms. For the latter, the surface dangling bonds are terminated by hydrogen atoms and the size of pore was fixed to 6 Å, which is the dominated pore size of the experimental samples. We used a $10 \times 10 \times 1$ k-point mesh for Brillouin zone sampling of the graphene electrode, while a single Γ-point was employed in the slit pore calculations. All calculations were carried out with the Quantum Espresso package[51].

**CV experiments**. HCAMs were synthesized following the procedure reported in our previous studies[32,52,53]. The dimensions of the electrodes were $2 \times 3$ cm with an average thickness of 470 μm. The electrode material has a mass density of 0.50 g $cm^{-3}$, with approximate BET and DFT surface area of 862.15 $m^2\,g^{-1}$ and 1084.167 $m^2\,g^{-1}$, respectively. Assuming a carbon density of 1.95 g $cm^{-3}$, the measured density implies an average porosity of 74%. CV experiments were conducted at room temperature using a BioLogic VSP-300 potentiostat. Specifically, two carbon aerogel electrodes were separated to prevent electrical short circuits using a 90-μm thick by $2.5 \times 3.5$ cm non-conductive polyester rectangular mesh, with an estimated porosity of 34%. In addition, homemade clippers with four titanium wires were used for contacts. The CV measurements were carried out for 0.05 and 1 M

solutions of NaCl, LiCl, KCl, and CsCl salts at three different scan rates of 1.5, 1.0, and 0.5 m Vs$^{-1}$. The capacitance of the carbon electrodes in the salt solutions was then calculated from CV experiments at different scan rates. In all cases, we chose the highest current value where there are no Faradaic reactions (between $-0.65$ and $-0.55$ V).

We also carried out capacitance measurements of 1 M LiCl and CsCl solutions using a three-electrode set-up, and we observed a similar trend compared to the two-electrode experiments (see Supplementary Methods). The three-electrode CV experiments were performed using a one-compartment cell consisting of a 3.8-mg small piece of carbon aerogel as the working electrode, a platinum wire as the counter electrode, and saturated calomel as the reference electrode; HOPG was used as the contact between the working electrode and current collector. The CV experiments of 1 M solutions of LiCl and CsCl were measured at six different scan rates, i.e., 50, 20, 10, 5, 2, and 1 m Vs$^{-1}$ (Supplementary Figs. 11 and 12). Capacitance for the different salts was calculated from CV experiments at different scan rates and is reported in Supplementary Fig. 13.

## Data availability

All data presented in this study are included in the article and Supplementary Information. The data are available from the corresponding authors upon request.

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

## Acknowledgement

This work was performed under the auspices of the U.S. Department of Energy by Lawrence Livermore National Laboratory under Contract DE-AC52-07NA27344. Financial support is from the Laboratory Directed Research and Development project (18-ERD-024). Computational support was from the LLNL Grand Challenge Program. The authors thank Liam Krauss for his critical reading of the manuscript.

## Author contributions

C.Z., T.A.P., B.C.W., M.S., and P.G.C. designed the research. C.Z. and T.A.P. performed first-principles calculations. M.R.C., S.A.H., and P.G.C. performed electrochemical measurements. M.O. implemented the DFT-RISM method in the Quantum Espresso code. C.Z. and T.A.P. wrote the manuscript with contributions from all other authors. All authors contributed to the discussion of the results.

## Competing interests

The authors declare no competing interests.
