## [Peer Review File · Nature Communications]

Reviewers' comments:

Reviewer #1 (Remarks to the Author):

Authors have carried out first-principles/continuum simulations to explore the adsorption behaviors of representative alkali-metal ions near graphitic interface. The effect of ion radius and pore size on the adsorption energy/charge transfer is revealed. We agree that accurately describing the interfacial structure is indeed important for interpreting the experimental observations. However, we feel that the as-obtained insights should be further clarified and reinforced. As a prime example, atomic insights to interpret the discrepancies between simulation and experimental studies have been proposed in previous simulation work (e.g., a novel kinetic-dominated charging mechanism in Ref. 6 in the main text), while this important and highly relevant issue is not well discussed in this work. A major revision is needed before possible publication on Nature Communications. Detailed comments:

Q1. Simulation model is suggested to be added into the main text for easy reading. In the method section, the equilibrium time is not provided, which is usually more than nanoseconds, e.g., > 8 ns (Ref. 23 and Ref. 24 in the main text).

Q2. In addition to the ionic radius, the relationship between adsorption energy and hydration radius/capacitive performance is recommended to be quantified.

Q3. The employed simulation methods have been proposed in previous work (Ref. 25 in the main text). Are there any advancement or modification on the methodology in this work?

Q4. For calculating the charge transfer, the Li⁺/Na⁺/K⁺/Cs⁺/Cl⁻ ions in the bulk electrolytes and confined space are described by neutral atoms or charged atoms with specific excess electron? e.g., Cl⁻ ion with 1 excess electron. Please specify the simulation setup on this.

Q5. In addition to the realistic description of interfacial structure, evaluating the electrolyte dynamics (e.g., diffusion coefficient) in simulations is also important for eliminating the discrepancies between simulation and experimental studies. For example, using an atomistic simulation, a novel kinetic-dominated charging mechanism that goes beyond traditional EDL theory and long-held axiom has been proposed to elaborate the atomic mechanism of EDL structure formation in great details, which well interprets the controversial ion effect in experiments (see Ref. 6 in the main text). From the point view of this work, it is suspected that ion kinetics play a crucial role in determining the charge storage mechanisms and interfacial structures, which is further demonstrated by electrochemical measurements, while the effect of dynamics is not included in this work. Authors are suggested to comprehensively comment on this important and highly relevant work.

Q6. Following the electrochemical measurements in Ref. 6, authors are recommended to explore the capacitance of ions at different scan rates (from 2 to 20 mV s⁻¹). A comprehensive comparison with Ref. 6 is also needed to evaluate the role of ion kinetics in determining the atomic charge storage mechanisms.

Q7. Authors stated that high amount of charge transfer will yield less voltage drop across the interface and therefore a larger capacitance. This trend should be quantified in the main text.

Q8. Can the as-obtained insights be extended to other electrolyte systems (e.g., water-in-salt electrolyte/ionic liquids) and non-graphitic electrode materials (e.g., RuO₂). A general consensus rather than specialized rule towards representative alkali-metal ions should be built, which can draw great interest for broad readers.

Reviewer #2 (Remarks to the Author):

This manuscript presents a combined computational and experimental study of the capacitance of atomistic-smooth graphite in contact with electrolyte solutions of different metal ions.

The computational study is done using a QM/continuum approach developed by one of the author. The data obtained from the simulations performed on the slit pore are then compared with experimental ones.

The overall conclusions of the manuscript is that the adsorption of ions onto graphitic surfaces is ion-dependent and is the results of a complex mixture of hydration free energy and surface

polarisation. This manuscript in particular proposes that charge transfer needs also to be taken into account to explain recent experimental capacitance data.

The topic of the manuscript is surely very timely and interesting. The increasing amount of experimental data of ions adsorption on graphitic surfaces show that this adsorption mechanism is much more complicated than expected and needs to be carefully understood to fully exploit this material for energy storage or membranes.

The manuscript is also well organised and sufficiently clear and I suggest publication after the authors address the following comments

1. I agree with the authors that classical fixed charged atomistic force fields cannot properly described the ion-dependence of the adsorption mechanism. The authors however include in this group of models also the work of Williams et al (ref 12) which however indicates the opposite and clearly proposes the idea of the ion-specificity of the adsorption mechanism. In that reference the authors cite also the importance of the hydration free energy and hydration shell size into the relative adsorption of ions as suggested also here by the authors. I ask the authors to make this distinction between classical models and commented on this previous work.
2. Said that the trend of adsorption published in Ref. 12 is different than that of the experimental work of Ref. 10 and that the authors explained (rather than predict with their model) proposing an extra effect: charge transfer. I would like the authors to comment on the importance of having the an electrified interface (rather than neutral) on the adsorption results. Do they expect that the charge transfer will not occur when the interface is not charged? Most of the membrane set ups are not charged.
3. Reference 10 (the experimental work) suggests that anion should be more adsorbed than cations. This seems to contradict the well known pi-cation interactions responsible for the ions adsorption. Did the authors see a similar trend?
4. On a more technical note: how many cations have been explicitly included in the DFT calculations? While the hybrid first principle/continuum method has been previously published, it would helpful for the reader to have summarised here (in the SI) the most important aspects and approximations made to fully appreciate the computational results.
5. I am surprised by the fact that Na⁺ and Li⁺ are not adsorbed on the surface for surface charge of -0.25V. These ions are adsorbed on uncharged graphene surface (see Ref 12 but also others) so why are not adsorbed at this negative surface potential? Fig 2S seems to indicate that Li⁺ is actually never adsorbed.
6. I find the use of the word "ion radius" confusing. Do the authors refer to the Stoke radius or the hydration radius? The two are different and if they refer to the latter it is basically the same than saying "hydration strength" so the sentence "the structure of the aqueous solutions at the interface with graphene is governed by both ion radius and hydration strength" sounds a repetition of the same effect.
7. Last point, the experiments in Ref 10 are done at very high concentration (6M). The simulations here seems to have been done at lower concentration (1M). Do the authors expect an effect of salt concentration?

Reviewer #3 (Remarks to the Author):

Review on the manuscript

Ions at graphitic interfaces: the role of charge transfer, ion hydration and confinement

by Cheng Zhang et al.

submitted to Nature Communications

The manuscript deals with the influence of monovalent cation types on the capacitance of carbon based electrical double layer capacitors (EDLC) with aqueous electrolytes. The main claim is that in addition to the applied voltage and the pore size, the capacitance is affected by the ion hydration and in particular also by the degree of charge transfer. The authors use a hybrid atomistic modelling approach combining first principles DFT with the RISM method to demonstrate systematic energetic trends for increasing cation size (Li⁺, Na⁺, K⁺, Cs⁺) near an ideal graphene surface, which they attribute - in addition to already previously discussed parameters such as hydration strength - to an increasing charge transfer with increasing ion size. Calculations for ions confined in a slit pore indicate that this effect may be even enhanced, predicting a systematic increase of capacitance from Li⁺ to Cs⁺. These modelling results are then compared with electrochemical measurements on a carbon aerogel sample, revealing qualitative agreement with the predictions.

Even though it might appear quite obvious that hydrated ions with essentially the same size as the confining pores should not be treated as simple point charges, the electronic structure of the ions has rarely been taken into account in this field. Therefore, I evaluate these results as fairly novel with a very high potential impact in the field, possibly going also beyond the presented scenario with implications for instance also for the understanding of biological systems. The manuscript is well presented and is in my view highly suitable for publication in Nature Communications. Having said this about the - admittedly more important - atomistic modelling part of the paper, I have some reservations concerning the presentation and possibly also the extent and quality of experimental data proposed for the validation of the predictions. Since the experimental system employed is far from the ideal case used in the atomistic models, the interpretation of the quite small experimental effects should be taken with caution. There may be many other effects influencing the capacitance to a much larger quantitative extent, e.g., a strong deviation of the real nanoporous carbon from the ideal graphene confinement, such as a complex percolating pore network with different pore sizes influencing both, the pore accessibility as well as the ion kinetics. Also the presence of surface functional groups different from H at defects or edges might influence the capacitance considerably stronger than the predicted effects. This might for instance be the reason, why also the opposite effect, i.e. an increasing capacitance with decreasing cation size for similar cations was reported (e.g. in Ref. 9 in the manuscript). In the following, I list a number of improvements that – beside of a more cautious discussion - should be considered before the manuscript can be published.

- 1) Fig. S4 shows the size distribution of the used aerogel sample. First of all, the statement in the caption that the pore size distribution (psd) was obtained from BET analysis is just nonsense; BET is generally not suitable to determine psd, and in addition, surface area determination by BET is not applicable to micropores. Authors should consider giving more information here (adsorption isotherm, way of determining the psd, assumptions, etc.). I guess the NLDFT method with slitpore assumption was employed (DFT is mentioned in the Methods section), since several apparent maxima in the psd are typical artifacts of this method. Such artifacts could be avoided by using for instance the QSDFT method. This would

also resolve the apparent contradiction of a maximum of the psd at 5 Å, where according to Figs. 3d and S6, the Cs⁺ ion should not be able to enter these pores.

- 2) CV measurements from full cells are presented for two concentrations at a slow scan rate of 0.5 mV/s. According to the Methods section they have been performed also for another two scan rates, but nothing is said about those measurements. I suggest showing all measurements either in a table or a plot of capacitance versus scan rate in the SI, ideally also for considerably higher scan rates to confirm the general trend of the capacitance. Moreover, I am wondering why not a 3-electrode setup was chosen that would allow determining the capacitance of a single electrode instead of the capacitance of the whole device. The authors should at least comment on these details in the SI.
- 3) In Fig. 2b a coordinate system should be given. Although it becomes clear from the text what the z-direction is, the discussion is somewhat hard to follow, since also the location of the ions with respect to the pore walls after intercalation is discussed.
- 4) I am somewhat puzzled by the Figure S3. If z is the distance between the (center of the) ion and the surface hydrogen, the numbers for z in the 6 panels cannot be correct. For instance, the upper right panel is denoted z=0, but the panel shows the ion fully intercalated into the slit pore. Again, a coordinate system (also in relation to Fig. 2b) should be given.

We would like to thank the Reviewers for their positive and valuable comments that allowed us to significantly improve the quality of our manuscript. Below are our responses to the Reviewers' comments.

Reviewer 1

1. *Simulation model is suggested to be added into the main text for easy reading. In the method section, the equilibrium time is not provided, which is usually more than nanoseconds, e.g., 8 ns (Ref. 23 and Ref. 24 in the main text).*

Author reply: In contrast with the computational approaches presented in Refs. 23-24 that rely on classical molecular dynamics simulations, our computational protocol is based on a recent developed hybrid solvation model (Ref. 25) for the description of solid-liquid interfaces. In this approach, the liquid components are described using the reference interaction site method (RISM), whereas the graphene electrode and explicit cations are treated by density functional theory (DFT) with the plane-wave basis and pseudopotentials technique. Since the solution was described by the solvation model, equilibration time is not relevant and dynamical effects are not considered (see further discussion below).

Following the Reviewer's suggestion, we have extended the Methods section to clarify the differences between hybrid first-principles/continuum simulations (DFT-RISM) employed in our study and the conventional approaches that rely on classical molecular dynamics simulations.

Page 13-14: Our computational methodology of aqueous solutions at graphitic interfaces relies on a newly developed approach that combines DFT with the effective screening medium (ESM) technique and implicit solvation model described in the framework of the reference interaction site method (RISM). While the DFT-RISM method has been employed to investigate the behavior of solvents at relatively simple solid-liquid interfaces, our study presents the first application of the approach to study solutions confined in nanopores, as well as the adsorption of an explicit ion at graphitic interfaces.

A schematic description of our hybrid quantum-continuum approach is shown in Figure S10. In particular, the explicit cations and carbon electrodes were described at the DFT level of theory, whereas the electrolyte was represented by an implicit solvent model consisting of water molecules and 1 M of salts at a temperature of 330 K. Atomic charges and Lennard-Jones (LJ) potentials of the solvent and ions are described through the OPLS all-atom force fields. The RISM calculations were performed with the closure model of Kovalenko and Hirata, with a cutoff energy of 300 Ry for the solvent correlation function. Our calculations of cations on charged electrodes were carried out by fixing the net charge of the explicit quantum mechanical region, which

is balanced by classical ions in the RISM solvation, and therefore the whole system is neutral. We stress that, in variation with classical molecular dynamics simulations that are often used to investigate solid/liquid interfaces, our calculations rely on a solvation model for the description of aqueous solutions. In this regard, dynamical effects are not considered in this work, and technical aspects such as the equilibration time in molecular dynamics simulations are not relevant.

2. *In addition to the ionic radius, the relationship between adsorption energy and hydration radius/capacitive performance is recommended to be quantified.*

Author reply: Following the Reviewer’s suggestion, we have provided detailed information of the ionic and hydrated radius of the alkali-metal ions, as well as their adsorption energies on graphene and the capacitance of the graphene slit-pores in different aqueous solutions in Table S1. We have also revised the manuscript to discuss in more detail the relationship between the ionic radius, adsorption energy, hydrated radius and capacitive performance.

Page 10: The cyclic voltammetry (CV) measurements of the HCAMs electrodes at two different solution concentrations of 50 mM and 1 M (Figure S6, Supporting Information) show that the capacitance follows the order $\text{Cs}^+ > \text{K}^+ > \text{Na}^+ > \text{Li}^+$ in aqueous solutions, with more pronounced trend at the higher concentration. The trend in the measured gravimetric capacitance, summarized in Figure 3a, is consistent with not only our theoretical predictions but also with previous experimental studies. In particular, it is shown that the capacitance increases with the ionic radius and the adsorption energy of the ions at the interface with graphene; however, it decreases with their hydration radius (see Table S1).

3. *The employed simulation methods have been proposed in previous work (Ref. 25 in the main text). Are there any advancement or modification on the methodology in this work?*

Author reply: Our simulation protocol is based on the DFT-RISM method proposed in Ref. 25 by one of our co-authors, Dr. Otani, for the description of solid-liquid interfaces. There is no modification of the methodology; however, we note that our study presents the first application of this approach to study solutions confined in nanopores and the adsorption of explicit ions at aqueous interfaces, which are far more complex compared to the systems presented in the original study. We have revised the manuscript to clarify the difference between the current study and Ref. 25.

Page 13: While the DFT-RISM method has been employed to investigate the behavior of solvents at relatively simple solid-liquid interfaces, our study presents the first application of the approach to study solutions confined in nanopores, as well as the adsorption of an explicit ion at graphitic interfaces.

4. *For calculating the charge transfer, the $\text{Li}^+/\text{Na}^+/\text{K}^+/\text{Cs}^+/\text{Cl}^-$ ions in the bulk electrolytes and confined space are described by neutral atoms or charged atoms with specific excess electron? e.g., Cl^- ion with 1 excess electron. Please specify the simulation setup on this.*

Author reply: For the potential energy surface calculation of an explicit cation near graphene (Figure 1a), the total net charge of the “graphene-cation” component that is explicitly treated by DFT is set to be $+0.5 e$, corresponding to a charge of $-0.5 e$ and $+1 e$ on the graphene and cation, respectively. For the confined systems, the total charge of the “pore-cation” was set as zero, corresponding to a charge of $-1.0 e$ and $+1 e$ on the slit-pore and cation, respectively. We also note that the net charge of the DFT region is balanced by the classical ions in the implicit solvation model through the variation of the classical ion density.

Following the Reviewer’s request, we have revised the manuscript to discuss in more

detail the simulation setup for the calculation of the charge transfer. We also added a Figure S10 in the Supporting Information to illustrate our simulation model.

Page 14: A schematic description of our hybrid quantum-continuum approach is shown in Figure S10. In particular, the explicit cations and carbon electrodes were described at the DFT level of theory, whereas the electrolyte was represented by an implicit solvent model consisting of water molecules and 1 M of salts at a temperature of 330 K. Atomic charges and Lennard-Jones (LJ) potentials of the solvent and ions are described through the OPLS all-atom force fields. The RISM calculations were performed with the closure model of Kovalenko and Hirata, with a cutoff energy of 300 Ry for the solvent correlation function. Our calculations of cations on charged electrodes were carried out by fixing the net charge of the explicit quantum mechanical region, which is balanced by classical ions in the RISM solvation, and therefore the whole system is neutral.

5. *In addition to the realistic description of interfacial structure, evaluating the electrolyte dynamics (e.g., diffusion coefficient) in simulations is also important for eliminating the discrepancies between simulation and experimental studies. For example, using an atomistic simulation, a novel kinetic-dominated charging mechanism that goes beyond traditional EDL theory and long-held axiom has been proposed to elaborate the atomic mechanism of EDL structure formation in great details, which well interprets the controversial ion effect in experiments (see Ref. 6 in the main text). From the point view of this work, it is suspected that ion kinetics play a crucial role in determining the charge storage mechanisms and interfacial structures, which is further demonstrated by electrochemical measurements, while the effect of dynamics is not included in this work. Authors are suggested to comprehensively comment on this important and highly relevant work.*

Author reply: We thank the Reviewer for their valuable comment. We agree that Ref. 6 provides significant understanding on the charging mechanism of the planar graphene electrode in aqueous solutions, and that dynamical effects can play important role in the capacitance between the ions. We have revised the manuscript to discuss in more detail the results obtained in Ref. 6 and how they are related to the current study. Additional discussion can also be found below in our response for Q6.

Page 4: As a prime example, Yang *et al.* calculated the capacitance of a graphene electrode in aqueous electrolytes with alkali-metal cations using classical MD simulations, showing that the capacitance does not depend on the cation type due to the high dielectric constant of liquid water. This is however in contrast to the experimental results reported in the same study. To explain this discrepancy, a kinetic-dominated charging mechanism was proposed and supported by capacitance measurements at different scan rates, pointing to the importance of ion kinetics on the electrochemical performance. However, electrochemical impedance spectroscopy (EIS) measurements in Ref. 11 show that distinctive difference in the capacitance between the cations remains at a low frequency; for instance, the measured capacitance in KCl and LiCl solutions were $\sim 8 \mu\text{F}/\text{cm}^2$ and $\sim 5 \mu\text{F}/\text{cm}^2$, respectively, resulting in a $C_{\text{K}^+}/C_{\text{Li}^+}$ ratio as large as 1.6. This indicates that ion kinetics may not be solely responsible for the difference in the capacitance between different aqueous solutions, motivating further development in theoretical descriptions of the electrical response of solutions at graphitic interfaces.

6. *Following the electrochemical measurements in Ref. 6, authors are recommended to explore the capacitance of ions at different scan rates (from 2 to 20 mV s⁻¹). A comprehensive comparison with Ref. 6 is also needed to evaluate the role of ion kinetics in determining the atomic charge storage mechanisms.*

Author reply: Following the Reviewer’s request, we have carried out additional ex-

periments to measure the capacitance at different scan rates. Specifically, as discussed further below in Q4 & Q5 of our Response to the Reviewer 3, we have carried out capacitance measurements for a scan rate up to 50 mV/s using a three-electrode setup. We find that, due to the resistive effects in activated carbon aerogel electrode systems and the relatively large thickness and mass of our material, CV measurements at high scan rates do not provide accurate capacitance information. Accordingly, we have reported the capacitance result only for the scan rate up to 1.5 mV s⁻¹; nevertheless, the overall behavior of the capacitance is consistent with the results presented in Ref. 6. In particular, we find that: (i) the capacitance of the hierarchical carbon aerogel monoliths (HCAMs) electrode in aqueous solutions decreases with increasing the scan rate; and (ii) the capacitance loss is more significant for ions with smaller ionic radius, e.g., Li⁺. These results are in agreement with the finding presented in Ref. 6, pointing to the importance of ion dynamics in determining the electrochemical performances.

We have added experimental results of the dependence of capacitance on the scan rate to the Supporting Information (Figures S8 & S9 and Table S2 & S3). We have also revised the manuscript to discuss the dynamical effects on the capacitance and comparison between our findings and that presented in Ref. 6.

Page 12: As a final note, it is important to comment on the role of ion dynamics, which is not considered in our simulations, in determining the capacitance of the graphitic electrodes. Along this direction, by using a combination of classical MD simulations and electrochemical measurements, Yang *et al.* concluded that the charging mechanism and capacitance of graphene in aqueous solutions with alkali-metal ions is dominated by the ion kinetics. Here, to elucidate the role of ion kinetics, we carried out capacitance measurements for different scan rates, and we find that the overall behavior of the capacitance is consistent with the results presented in Ref. 6. In particular, we find that the capacitance of the HCAMs electrode in aqueous solutions decreases with increasing the scan rate, and that the capacitance loss is more significant for ions with a smaller ionic radius (see Figures S8 & S9 and Tables S2 & S3). For instance, we find the capacitance decrease of LiCl is higher than CsCl with increased scan rates, which can be associated with a slower diffusion of Li⁺. These results indicate that, together with specific ion effects and confinement, ion dynamics may play additional roles in determining the electrochemical performances of graphitic electrodes in aqueous solutions.

7. *Authors stated that high amount of charge transfer will yield less voltage drop across the interface and therefore a larger capacitance. This trend should be quantified in the main text.*

Author reply: Following the Reviewer’s suggestion, we have revised the manuscript to provide a more quantitative description of how the voltage drop depends the type of cation.

Page 8: When charge transfer from graphene to the cations is taken into account, not all of the excess charge is diverted towards filling the electrode electronic states; rather, a part of it is transferred to the electrolyte, leading to a smaller ΔV_{total} . This implies that electrolytes that experience stronger charge transfer with graphene would experience less voltage drop across the interface and therefore yield a larger capacitance. For instance, we find a Fermi level shift in graphene (labeled ΔV_{CT} in Figure 1c) of 0.27 eV, 0.31 eV, 0.40 eV and 0.42 eV for Li⁺, Na⁺, K⁺ and Cs⁺, respectively, when the ions approach the interface from a distance of 6 Å to 2 Å.

8. *Can the as-obtained insights be extended to other electrolyte systems (e.g., water-in-salt electrolyte/ionic liquids) and non-graphic electrode materials (e.g., RuO₂). A general consensus rather than specialized rule towards representative alkali-metal ions should*

be built, which can draw great interest for broad readers.

Author reply: Following the Reviewer’s suggestion, we have revised the manuscript to discuss how our findings are relevant to other electrolytes and non-graphitic electrode materials.

Page 13: Although our studies were carried out for carbon materials and aqueous solutions, the findings presented here, particularly on specific ion effects, charge transfer and their interplay with confinement, are likely to hold for other systems with different electrolytes, such as ionic liquids and organic electrolytes, and non-graphitic electrodes, such as low-dimensional systems, e.g., MXene and MoS₂. Finally, we note that the computational strategy based on the DFT-RISM approach presented in this work is general and applicable to other electrochemical interfaces.

Reviewer 2

1. *I agree with the authors that classical fixed charged atomistic force fields cannot properly described the ion-dependence of the adsorption mechanism. The authors however include in this group of models also the work of Williams et al. (Ref. 12) which however indicates the opposite and clearly proposes the idea of the ion-specificity of the adsorption mechanism. In that reference the authors cite also the importance of the hydration free energy and hydration shell size into the relative adsorption of ions as suggested also here by the authors. I ask the authors to make this distinction between classical models and commented on this previous work.*

Author reply: We thank the Reviewer for their suggestion. In Ref. 12, Williams *et al.* parametrized the ion-carbon interaction using a combination of density functional theory and the conductor-like polarizable continuum model (CPCM) for liquid water. Using these force fields, their simulations of 1 M electrolyte solutions show that cations are strongly adsorbed onto the graphene surface with a trend $\text{Li}^+ < \text{Na}^+ < \text{K}^+$. In addition, the authors show that, in the presence of liquid water, these ions adsorb on a graphene flake with a binding energy of -10.4 kJ/mol, -13.8 kJ/mol and -12.6 kJ/mol for Li^+ , Na^+ and K^+ , respectively.

Comparing our simulation results to Ref. 12, the absolute adsorption energy of the cations on graphene show some discrepancies. In particular, in contrast to Ref. 12, we find that Li^+ and Na^+ are not adsorbed on the graphene surface. However, we stress that the overall trend is consistent between the two studies; specifically, in agreement with our calculations, Ref. 12 shows that cations with a larger ionic radius and a weaker hydration solvation structure tends to adsorb stronger with graphene.

We believe that the discrepancy in the absolute value of the binding energies of cation on graphene between our calculations and Ref. 12 stems from several technical details associated with the solvation models employed in the two studies. In particular, in Ref. 12, the cation adsorption energy is calculated for the pure water described by the CPCM model, whereas our calculations were carried out for 1 M electrolytes, which included ion effects. As we show in Figure S3, Supporting Information, the inclusion of classical ions in our simulation could weaken the interaction between a charged graphene and explicit cation. In addition, comparing with CPCM, the RISM approach employed in our study allows for the description of the local solvent structure around the ions. In this regard, the RISM approach provides a more realistic description of the desolvation process of ions when they approach the graphene surface, which introduces additional energy penalty for ion adsorption. Collectively, these differences in the solvation models and computational setup explain for the discrepancy in the binding energy between our study and Ref. 12.

Following the Reviewer’s request, we have revised the manuscript to clarify the differ-

ence in the computational method employed in the current study and that of Ref. 12.

Page 6: The conclusion that cation with a larger ionic radius exhibits stronger adsorption at the interface with graphene is also consistent with a recent theoretical study using a combination of DFT and the conductor-like polarizable continuum model (CPCM) for liquid water. However, it is necessary to point out that the absolute binding energy obtained for Li^+ , Na^+ and K^+ on graphene is weaker in the current study compared to that of Ref. 12, which can be attributed to several differences in the computational setup and solvation models. In particular, in Ref. 12, the cation adsorption energy is calculated for pure water described by the CPCM model, whereas our calculations were carried out for 1 M electrolytes. As we show in Figure S3, Supporting Information, the inclusion of ions in our simulation weakens the interaction between a charged graphene and cations. In addition, in variation with the CPCM model, the RISM approach employed here allows for the description of the local solvent structure around the ions, as illustrated in the inset of Figure 1a and Figure S10. In this regard, the RISM approach provides a more realistic description of the desolvation process of ions when they approach the graphene surface, which introduces additional energy penalty for ion adsorption.

2. *Said that the trend of adsorption published in Ref. 12 is different than that of the experimental work of Ref. 10 and that the authors explained (rather than predict with their model) proposing an extra effect: charge transfer. I would like the authors to comment on the importance of having the an electrified interface (rather than neutral) on the adsorption results. Do they expect that the charge transfer will not occur when the interface is not charged? Most of the membrane set ups are not charged.*

Author reply: As shown in Figure 1a and Figure S2, our study reports the potential energy surface of cation adsorption on graphene for different charge of the electrode. As expected, we find that the binding energy between the cations and graphene is stronger for a more negatively charged graphene, resulting in a charge-dependent adsorption energy of the cations. To clarify this point, we have added Table S1 to the Supporting Information to report the binding energy between the cations and graphene with different net charge.

For either charged or neutral surfaces, as we already show in the current version of the manuscript, the charge transfer always exists due to the cation’s polarization on the graphene surface. Specifically, it is shown in Figure 1b that charge transfer occurs on both neutral and charge surface, and the degree of the charge transfer largely depends on the cation type and ion-graphene distance.

Page 6: We note that adsorption of these smaller cations can be induced by further increasing the electrode charge density, eventually leading to binding for Li^+ and Na^+ at 2.3 Å and 2.7 Å from the graphene plane, respectively (see Figure S2 and Table S1, Supporting Information).

3. *Reference 10 (the experimental work) suggests that anion should be more adsorbed than cations. This seems to contradict the well known π -cation interactions responsible for the ions adsorption. Did the authors see a similar trend?*

Author reply: The comparison between anion and cation adsorption in Ref. 10 was shown in Figure 3 for LiCl, where the authors find that Cl^- adsorption at the positive potential is stronger than Li^+ adsorption at the negative potential. Using the same computational methodology presented in the current study, we find that the Cl^- -graphene interaction is weaker than most of the cations (Na^+ , K^+ , Cs^+), but stronger than Li^+ (see the attached Figure for Review only). This is consistent with the results presented in Figure 3 in Ref. 10 for LiCl. More detailed analysis of anion adsorption and comparison between anions and cations behavior at the graphitic interface will be provided in our future publication.

4. *On a more technical note: how many cations have been explicitly included in the DFT calculations? While the hybrid first principle/continuum method has been previously published, it would be helpful for the reader to have summarised here (in the SI) the most important aspects and approximations made to fully appreciate the computational results.*

Author reply: For all the simulations of cation adsorption, we included one explicit cation in the simulation box, while other ions were described by classical model within the RISM framework to match the bulk concentration of 1 M (see Figure S10). For instance, when one explicit Li^+ is adsorbed on a neutral graphene, the number of classical ions determined by the RISM in our calculations is 7 and 6 for Cl^- and Li^+ , respectively, in the electrolyte region. We have added Figure S10 in the Supporting Information to clarify the most important aspects and physical approximations in our hybrid DFT/continuum simulation.

5. *I am surprised by the fact that Na^+ and Li^+ are not adsorbed on the surface for surface charge of -0.25V . These ions are adsorbed on uncharged graphene surface (see Ref. 12 but also others) so why are not adsorbed at this negative surface potential? Fig 2S seems to indicate that Li^+ is actually never adsorbed.*

Author reply: Our simulation indicated that Li^+ and Na^+ are less prone to adsorb on neutral or weakly charged graphene surface compared to the results reported in Ref. 12. As already discussed above (Q1), we believe that this discrepancy stems from the differences in the solvation models employed in the two studies. Specifically, in variation with the CPCM model, the RISM approach employed here allows for the description of the change in the local solvent structure around the ions. In this regard, the RISM approach provides a more realistic description of the desolvation process of ions when they approach the graphene surface, which in turn introduces additional energy penalty for ion adsorption.

6. *I find the use of the word ion radius confusing. Do the authors refer to the Stoke radius or the hydration radius? The two are different and if they refer to the latter it is basically the same than saying hydration strength so the sentence solutions at the interface with graphene is governed by both ion radius and hydration strength sounds a repetition of the same effect.*

Author reply: We thank the Reviewer for their comment. We have changed all the “ion radius” in the manuscript to “ionic radius”, which refers to the bare ion size.

Page 7: Accordingly, we conclude that the structure of aqueous solutions at the interface with graphene is governed by both ionic radius and hydration strength, as well as applied potential, which alters the relative competition between these two factors.

7. *Last point, the experiments in Ref. 10 are done at very high concentration (6M). The simulations here seem to have been done at lower concentration (1M). Do the authors expect an effect of salt concentration?*

Author reply: We believe that our conclusion is not affected by the ion concentration. For instance, our experimental results for capacitance show the same trend for 50 mM and 1.0 M aqueous solutions. We also note that our calculations were carried out at 1 M, which is not far from the concentration of the main experimental data reported in Ref. 10 (Figure 3), where the authors compare the capacitance of different 0.5 M electrolytes. We have revised the manuscript to clarify this point.

Page 10: The cyclic voltammetry (CV) measurements of the HCAMs electrodes at two different solution concentrations of 50 mM and 1 M (Figure S6, Supporting Information) show that the capacitance follows the order $\text{Cs}^+ > \text{K}^+ > \text{Na}^+ > \text{Li}^+$ in aqueous solutions, with more pronounced trend at the higher concentration.

Reviewer 3

1. *Since the experimental system employed is far from the ideal case used in the atomistic models, the interpretation of the quite small experimental effects should be taken with caution. There may be many other effects influencing the capacitance to a much larger quantitative extent, e.g., a strong deviation of the real nanoporous carbon from the ideal graphene confinement, such as a complex percolating pore network with different pore sizes influencing both, the pore accessibility as well as the ion kinetics. Also the presence of surface functional groups different from H at defects or edges might influence the capacitance considerably stronger than the predicted effects. This might for instance be the reason, why also the opposite effect, i.e. an increasing capacitance with decreasing cation size for similar cations was reported (e.g. in Ref. 9 in the manuscript).*

Author reply: We thank the Reviewer for their constructive comment. We have revised the manuscript to discuss other possible effects that may influence the measured capacitance of porous carbons.

Page 12: As a final note, it is important to comment on the role of ion dynamics, which is not considered in our simulations, in determining the capacitance of the graphitic electrodes. Along this direction, by using a combination of classical MD simulations and electrochemical measurements, Yang *et al.* concluded that the charging mechanism and capacitance of graphene in aqueous solutions with alkali-metal ions is dominated by the ion kinetics. Here, to elucidate the role of ion kinetics, we carried out capacitance measurements for different scan rates, and we find that the overall behavior of the capacitance is consistent with the results presented in Ref. 6. In particular, we find that the capacitance of the HCAMs electrode in aqueous solutions decreases with the scan rate, and that the capacitance loss is more significant for ions with smaller ionic radius (see Figures S8 & S9 and Table S2 & S3). For instance, we find the capacitance decrease of LiCl is stronger than CsCl with increased scan rates, which can be attributed to a slower diffusion of Li⁺ compared to Cs⁺. Collectively these results indicate that, together with specific ion effects and confinement, ion dynamics may play additional roles in determining the electrochemical performances of graphitic electrodes in aqueous solutions. Besides the intrinsic effects of ion kinetics, ion transport in porous carbons can also be affected by the deviation of the electrode material from the ideal graphene slit-pore due to, e.g., a complex percolating pore network with different pore sizes. Finally, the presence of functional groups or defects in the electrode may also introduce additional effects on the capacitive performance.

2. *Fig. S4 shows the size distribution of the used aerogel sample. First of all, the statement in the caption that the pore size distribution (psd) was obtained from BET analysis is just nonsense; BET is generally not suitable to determine psd, and in addition, surface area determination by BET is not applicable to micropores. Authors should consider giving more information here (adsorption isotherm, way of determining the psd, assumptions, etc.).*

Author reply: We thank the Reviewer for pointing out this mistake. We have revised the Supporting Information to provide more detailed information on how the pore size distribution is measured and determined.

Figure S5, Supporting Information: Pore size characteristics of the hierarchical carbon aerogel monoliths (HCAMs), including volume derivative with respect to the pore width, and cumulative pore volume. To measure the micro-porosity, we carried out N₂ absorption at 77 K with a Micromeritics ASAP 2020. In addition, in order to adequately resolve the micropore structure, the experiment was conducted over a 100 h time period to allow the N₂ time to diffuse into the narrow porous regions. The final pore size distribution (PSD) was obtained using both the NLDFT Standard Slit

model and the 2D-NLDFT Heterogeneous Surface model processed in the SAIEUS software package. The resulting PSD plots show substantial similarities between the two models, with the key feature being the majority of pore volume located at $\sim 5\text{-}6$ Å.

3. *I guess the NLDFT method with slitpore assumption was employed (DFT is mentioned in the Methods section), since several apparent maxima in the psd are typical artifacts of this method. Such artifacts could be avoided by using for instance the QSDFT method. This would also resolve the apparent contradiction of a maximum of the psd at 5 Å, where according to Figs. 3d and S6, the Cs⁺ ion should not be able to enter these pores.*

Author reply: We thank the Reviewer for their suggestion. Our analysis of the pore size distribution (PSD) was carried out using the NLDFT method with infinite slit pore model. Due to the limited models that are available in our instrument, we were not able to fit the pore size distribution using the QSDFT method. We have instead employed different available NLDFT models to refit the PSD. As shown in Figure S5, the PSD obtained for different models exhibits qualitative agreement in the peak position, with an uncertainty of less than 1.0 Å for the main peak between 5-6 Å.

We note that an exact determination of the PSD is certainly beyond the scope of our paper. The analysis also shows that, since the PSD depends on the model employed, the true pore size might be slightly larger than 5 Å, and this might explain for the intercalation of Cs⁺ into the pores.

Page 11: It is necessary to point out the porosimetry measurements performed using gas sorption on dry HCAM electrodes indicate that the pore size is dominated by 5 Å pores (grey line in Figure 3d and black line in Figure S5, Supporting Information), which would seem to limit Cs⁺ adsorption and insertion according to the predictions. However, during CV cycling, the wetted electrodes are clearly capable of ion adsorption and exhibit large capacitance, as predicted for slightly larger pore geometries. This suggests a possible increase in the pore size upon wetting under operation, or that the true pore size of the dry HCAM electrodes is slightly larger than 5 Å due to the uncertainty introduced by the models employed for the determining of pore size distribution (see Figure S5).

4. *CV measurements from full cells are presented for two concentrations at a slow scan rate of 0.5 mV/s. According to the Methods section they have been performed also for another two scan rates, but nothing is said about those measurements. I suggest showing all measurements either in a table or a plot of capacitance versus scan rate in the SI, ideally also for considerably higher scan rates to confirm the general trend of the capacitance.*

Author reply: Following the Reviewer's suggestion, we have added Tables S2 & S3 and Figures S8 & S9 to the Supporting Information to summarize the capacitance measurement at different scan rates. Our results show consistent trend of capacitance for different cations at all scan rates. In addition, we have carried out capacitance measurements at considerably higher scan rates for the three-electrode setup (Figure S13, also see Q5 below) for the two representative solutions, i.e., LiCl and CsCl, and we reached the same conclusion.

5. *Moreover, I am wondering why not a 3-electrode setup was chosen that would allow determining the capacitance of a single electrode instead of the capacitance of the whole device. The authors should at least comment on these details in the SI.*

Author reply: Our experimental measurements were carried out using a two-electrode setup, which has been used in Ref. 7 for investigating the capacitance of graphene in aqueous solutions. Following Reviewer's suggestion, we have measured the CVs of 1 M LiCl and CsCl at different scan rates using a three-electrode set up, and observed similar trend compared to the two-electrodes experiments. We have added a section

and Figures S11-13 to the Supporting Information to discuss the results obtained with a three-electrode setup. We have also revised the manuscript accordingly.

Page 15: We also carried out capacitance measurements of 1 M LiCl and CsCl solutions using a three-electrode setup, and we observed similar trend compared to the two-electrodes experiments (see Section on three-electrode measurements, Supporting Information.)

Page 9, Supporting Information: Three-electrodes *CV* experiments were performed using a one compartment cell consisting of a 3.80 mg small piece of carbon aerogel as the working electrode, a platinum wire as the counter electrode and saturated calomel as the reference electrode (SCE), highly ordered pyrolytic graphite (HOPG) was used as the contact between the working electrode and current collector. *CV* experiments of 1 M solutions of LiCl and CsCl were measured at six different scan rates 50, 20, 10, 5, 2 and 1 mV/s (Figure S11 and S12). Capacitance for the different salts was calculated from *CV* experiments at different scan rate (Figure S13), in all cases we chose the highest current value where there are not faradaic reactions occurring.

Due to the high resistant in thick monolithic activated HCAM electrode systems, measuring cyclic voltammetry at higher scan rates provides less accurate capacitance information. The *CV* of 1 M LiCl and CsCl at different scan rates was measured using a three-electrode set up, particularly at scan rates below 10 mV/s where more accurate capacitance values can be determined, and we observed similar trend compared to the two-electrodes experiments.

We also note that the calculated capacitance with the three-electrodes is lower than the one obtained with the two-electrode set up, this can be explained by the higher resistance observed in the three-electrode measurements, due to poor contact between our thick monolithic electrode and the current collector, as well as the differences in the electrode geometry of both methods. Most importantly, the calculated capacitance with the two electrode measurements accounts for both the cation and the anion, while the calculated capacitance with the three-electrode measurements accounts for the cation or the anions depending on the potential polarity. The values plotted on Figure S13 is only for the cation ions.

6. *In Fig. 2b a coordinate system should be given. Although it becomes clear from the text what the z-direction is, the discussion is somewhat hard to follow, since also the location of the ions with respect to the pore walls after intercalation is discussed.*

Author reply: Following the Reviewer's comment, we have revised the Figure 2 and the caption to clarify the intercalation direction.

7. *I am somewhat puzzled by the Figure S3. If z is the distance between the (center of the) ion and the surface hydrogen, the numbers for z in the 6 panels cannot be correct. For instance, the upper right panel is denoted $z = 0$, but the panel shows the ion fully intercalated into the slit pore. Again, a coordinate system (also in relation to Fig. 2b) should be given.*

Author reply: Following the Reviewer's comment, we have revised the Figure S4 and the caption to clarify the coordinate system. In our simulation, $z = 0$ was chosen as the position associated with the configuration of dehydrated cations inside the pore.

REVIEWERS' COMMENTS:

Reviewer #1 (Remarks to the Author):

The concerns have been addressed. Additional experimental results have been added to support the statements on ion transport. The only suggestion is:

Please revise 'Yang et al.' to 'Bo et al.' in pages 4 and 12. It is better to use the name of corresponding author.

Reviewer #2 (Remarks to the Author):

I am satisfied with the changes and explanations of the authors to my remarks and I recommend publication of the manuscript as it is.

Reviewer #3 (Remarks to the Author):

All my comments and suggestions from the previous review have been properly and satisfactorily addressed. Therefore, from my side the paper can be published in Nature Communications without further changes.

Just for information to the authors; in the meanwhile an interesting paper (<https://doi.org/10.1021/acsaem.9b01069>) has been published, addressing the influence of percolating pore networks on capacitance. This paper might be cited along with the corresponding discussion in the manuscript.

We would like to thank the Reviewers for their valuable comments. Below are our responses to the Reviewers' comments.

Reviewer 1

1. *Please revise 'Yang et al.' to 'Bo et al.' in pages 4 and 12. It is better to use the name of corresponding author.*

Author reply: Following the Reviewer's suggestion, we have revised the manuscript accordingly.

Reviewer 3

1. *Just for information to the authors; in the meanwhile an interesting paper has been published, addressing the influence of percolating pore networks on capacitance. This paper might be cited along with the corresponding discussion in the manuscript.*

Author reply: We thank the Reviewer for their suggestion. We have cited the suggested paper in the revised manuscript.